# Financial cost of assisted reproductive technology for patients in high-income countries: A systematic review protocol

Purity Njagi[1]*, Wim Groot[1,2], Jelena Arsenijevic[3], Gitau Mburu[4], Georgina Chambers[5‡], Carlos Calhaz-Jorge[6‡], James Kiarie[4]

1 UNU-Maastricht Economic and Social Research Institute on Innovation and Technology, Maastricht University, Maastricht, The Netherlands, 2 Department of Health Services Research, Faculty of Health, Medicine and Life Sciences, Maastricht University, Maastricht, The Netherlands, 3 School of Governance, Faculty of Law, Economics and Governance, Utrecht University, Utrecht, The Netherlands, 4 Department of Sexual and Reproductive Health and Research, World Health Organization, Genève, Switzerland, 5 National Perinatal Epidemiology and Statistics Unit (NPESU), Centre for Big Data Research in Health and School of Clinical Medicine, University of New South Wales (UNSW), Sydney, Australia, 6 Clinic of Obstetrics and Gynaecology, Faculdade de Medicina da Universidade de Lisboa, Lisbon, Portugal

☉ These authors contributed equally to this work.
‡ GC and CCJ also contributed equally to this work.
* purity.njagi@maastrichtuniversity.nl, puritynjagih@outlook.com

**Data Availability Statement:** No datasets were generated or analysed during the current study. All

## Abstract

### Background

Infertility affects one in six people globally, with similar prevalence rates across high-income and low- and middle-income countries. Technological advancements, particularly in Assisted Reproductive Technology (ART), have improved fertility treatment options. Although access to ART is presumed to be better in high-income countries (HICs), economic factors and eligibility restrictions could still impact effective utilization in these settings. Informed by the Preferred Reporting Items for Systematic Reviews and Meta-Analyses protocols (PRISMA-P), this protocol outlines the methodological and analytical approaches to examine the ART costs paid by patients in HICs and the correlation with economic indicators and ART regulatory frameworks.

### Methods

Following the PRISMA approach, we will search for articles indexed in PubMed, EMBASE, Cumulative Index of Nursing and Allied Health Literature (CINAHL), Web of Science, PsycINFO, and Latin American & Caribbean Health Sciences Literature (LILACS). Grey literature from relevant organizations' virtual databases will also be searched. The review will encompass studies published between 2001 and 2024, with the primary outcome being ART direct medical and direct non-medical costs, while secondary outcomes shall include ART financing arrangements. The review will synthesize ART costs, adjusting them to USD Purchasing Power Parity for cross-country comparison, and correlate findings with GNI per capita and ART financing policies. The Integrated Quality Criteria for Review of Multiple

relevant data from this study will be made available upon study completion.

**Funding:** This work has received funding from the UNDP-UNFPA-UNICEF-WHO-World Bank Special Programme of Research, Development and Research Training in Human Reproduction (HRP), a cosponsored programme executed by the World Health Organization (WHO). Grant Number: APW-PO 203363893.

**Competing interests:** The authors have declared that no competing interests exist.

Study Designs (ICROMS) tool will be utilized to evaluate the quality of the included studies. We will conduct a meta-analysis if the studies provide sufficient cost-effect size estimates.

## Discussion

The review findings will contribute to our understanding of the potential financial burden faced by (disadvantaged) individuals in HICs due to ART costs. Additionally, the review shall highlight the implications that ART financing policies have in enhancing access and affordability, offering valuable insights for healthcare planning and policy formulation. The results will be disseminated through a peer-reviewed journal article and relevant international conferences.

## Trial registration

**Systematic review registration:** PROSPERO number: CRD42023487655.

## Introduction

Infertility is a prevalent global public health concern affecting millions of people. The World Health Organization (WHO) estimates that one in six people experience infertility at some point in their lives [1].The prevalence of lifetime infertility varies, ranging from 12.6% to 17.5%, with no significant differences between high-income countries (HICs) and low and middle-income countries (LMICs) [2].Over the last decades, technological advancements have greatly improved fertility treatment options, enabling many individuals to achieve pregnancy [3]. However, the availability of Assisted Reproduction Technology (ART) varies worldwide, with HICs offering more facilities and access to a broader range of advanced options. The impact of ART in these settings is evidenced by the increasing number of reported treatments, variation in treatment options, and a rising contribution to births [4, 5].

Despite technological options in ART, its costs remain high from an individual perspective, and significant barriers to cost reduction persist due to the need for highly skilled personnel, advanced equipment and for-profit service delivery in many countries [6]. Nevertheless, the presence of public funding in many HICs' healthcare systems has improved ART accessibility, resulting in an increased number of countries offering treatment, more clinics providing ART, and more individuals receiving treatment [7].

Despite this increase, access and affordability are not uniform among HICs or across all segments of the population in these countries, for several reasons. First, the economic status of a country plays a significant role in determining the availability and accessibility of ART, and ART utilization has been shown to be positively correlated with national wealth (GDP). For instance, Dong (2022) and Lass (2019) have demonstrated a positive correlation between ART utilization and GDP, and a negative correlation between ART success rates and GDP, in both HICs and LMICs [8, 9]. Furthermore, according to recent findings from LMICs, there is an inverse relationship between ART costs and both GDP per capita and average annual income, with affordability of ART being lower in poorer countries [10].

Second, although public funding or third-party insurance is provided in many HICs, the extent of ART coverage varies [11]. Limits on the number of in vitro fertilization (IVF) cycles funded by public or private subsidies exist based on factors such as female age, personal income, existence of previous children, BMI, type of couple, and the duration of infertility [11,

12]. Due to these limitations, patients have to pay Out-of-Pocket (OOP) if they do not meet the eligibility criteria, have exhausted the number of funded cycles, or when co-payments are required. Additionally, some countries, such as Australia, Austria, Chile, Germany, Denmark, Ireland, and Italy provide partial coverage or reimbursements for multiple cycles, while others, like Canada, Chile, New Zealand, Romania, and the United Arab Emirates, provide full coverage for a single cycle [11]. Thus, even in HICs the cost can be a potential barrier to ART access, especially for low-income individuals who require additional cycles beyond what is publicly funded. For instance, a study in Norway shows that despite the partial subsidization of ART services, individuals contribute about 3% of the average annual gross income for three cycles [13] leading to financial burden for the most disadvantaged.

The financial barrier is evident in patients' decisions to not seek consultations or treatment due to OOP payments [14, 15]. Furthermore, in countries like the United States of America (USA), and Japan, household income has been associated with the cost of fertility care, which has particularly influenced the type and intensity of treatment sought [16, 17]. An econometrics analysis of 30 countries demonstrated that, the cost of ART treatment in relation to annual disposable income is independently associated with ART utilization. Consequently, higher (average) incomes, government reimbursements and third party insurance can increase utilization [18]. For instance, in the USA, state-mandated insurance coverage for IVF has been shown to result in greater utilization [19]. Nonetheless, disparities in access exist due to variations in qualifications for coverage, the extent of coverage, and exemptions across the mandated states [20].

Third, restrictions on public financing have created disparities in access within and across countries, prompting a growing trend in the number of individuals seeking services from the private sector and Cross Border Reproductive Care (CBRC). For example, in 2016, in the United Kingdom (UK), 59% of IVFs were privately funded outside of the National Health Service (NHS) [21]. Additionally, in the 2021 International Federation of Fertility Societies (IFFS) survey 71% of respondents (n = 73) reported that individuals travel to their country to obtain ART at lower cost [11]. Failure to meet the pre-requisite ART coverage criteria and legal restrictions such as the type of treatment allowed in the home country have contributed to patients traveling abroad for ART [22]. Other factors that drive CBRC include lower costs, access to advanced techniques, perceived higher-quality treatments, privacy, and more permissive legal and ethical restrictions in destination countries [12, 23]. Nonetheless, CBRC involves extra expenses beyond medical care, including costs for travel and accommodation in the destination countries. Moreover, CBRC entails financial issues related to the repatriation of offspring and the management of complications during and after ART [24].

Finally, measurement methodologies vary and can influence the perceived cost dynamics. Several studies have examined the cost of ART using a societal, national, or government perspective and the implications for the healthcare system in HICs [25, 26]. Others have concentrated on the cost-effectiveness of various treatments per live birth [27–29]. Additionally, while some researchers assess the cost of ART per se, others assess both the cost of ART procedures as well as their downstream consequences. For example, Crawford *et al.* and Peeraer *et al* identified cost differences between single and multiple embryo transfers [30, 31], due to costs associated with the management of multiple pregnancies. Thus, the cost of ART from a patient perspective could be influenced by the underlying cost of healthcare systems, and the degree of financial support from both public sources and third-party payers [32, 33].

Given the above, analyzing the economic aspects of ART requires careful consideration of the approach taken to cost assessment. For instance, some studies have analyzed ART cost variations, including OOP costs, those borne by government and third-party insurance [32, 33]. These studies compared ART costs by converting local currency costs into US dollars and

Euros using interbank exchange rates. However, the choice of currency conversion method can impact the results and their interpretation. Market exchange rates are not ideal for sectors like healthcare since they are determined by the supply and demand for currencies. In contrast, using Purchasing Power Parities (PPPs) allows results to be valued at a consistent price level, reflecting only the differences in the volumes of goods and services consumed [34]. For instance, PPP has been previously applied to assess the impact of consumer cost on ART utilization [18]. This review will apply PPP to account for relative economic differences between countries, thereby providing more accurate cost estimates for meaningful cross-country comparisons. In addition, this review will further correlate ART costs with the GNI per capita, average income and ART regulatory frameworks such as financing and eligibility criteria to provide insights into the factors that can enhance affordability and universal access to ART.

## Objective

The primary objective of this systematic review is to appraise and synthesize the available evidence on the cost of ART to patients in high-income countries (HICs) and to establish the correlation with economic indicators and ART regulatory frameworks.

Specifically, the review aims to:

a. Determine the direct medical and direct non-medical costs of ART incurred by patients in HICs, with a distinction between OOP, costs absorbed through government subsidization, reimbursements, or third-party insurance.

b. Explore the variations in the cost of ART borne by patients across HICs adjusting for purchasing power parity (USD PPP).

c. Assess the affordability of ART across countries by comparing the costs by GNI per capita (USD PPP), average income, and minimum wage.

d. Examine the correlation between the ART costs borne by patients across the HICs with ART regulatory frameworks, including financing and eligibility criteria.

## Materials and methods

### Study design

This review protocol is consistent with the guidelines outlined in the Preferred Reporting Items for Systematic Review and Meta-Analysis Protocols (PRISMA-P) [35] (S1 Checklist). The systematic review process and reporting will follow the standards outlined in the Preferred Reporting Items for Systematic Reviews and Meta-Analyses (PRISMA) statement [36].

This protocol is registered in the PROSPERO database (http://www.crd.york.ac.uk/PROSPERO/) with the registration number CRD42023487655 to eliminate any potential duplication-related redundancies. Any deviations from this protocol shall be disclosed and explained to minimize reporting bias and maintain transparency.

### Search strategy

A systematic literature search strategy will be developed and implemented in collaboration with an academic librarian who will assist in developing search strings, refining search terms, optimizing and piloting the search strategy to ensure a comprehensive search of the relevant literature. The following electronic medical and health databases shall be searched: PubMed, EMBASE, Cumulative Index of Nursing and Allied Health Literature (CINAHL), Web of Science, PsycINFO, and Latin American & Caribbean Health Sciences Literature (LILACS).

The search of databases will be supplemented by a search of the grey literature through Google Scholar and online libraries of relevant organizations such as the World Health Organization (WHO), the World Bank, the International Federation of Gynecology and Obstetrics (FIGO), the International Federation of Fertility Societies (IFFS), and the International Committee for Monitoring Assisted Reproductive Technologies (ICMART).

Other databases include the European Society of Human Reproduction and Embryology (ESHRE), the American Society for Reproductive Medicine (ASRM), the Latin American Network of Assisted Reproduction (REDLARA), The Asia & Oceania Federation of Obstetrics and Gynaecology (AOFOG), and the Asia Pacific Initiative on Reproduction (ASPIRE). In addition, related conference proceedings and abstracts will be searched.

## Search string of keywords

The search strategy will use a combination of keywords and Medical Subject Headings (MeSH) related to infertility treatment and cost in HICs. The search strategy will be adapted and tailored for each database, as appropriate, using a variety of approaches, including Boolean operators, truncations, proximity operators, and Subject Headings, among others. The provisional search strings for the databases are provided in S1 File.

The search terms shall include:

1. 'Costs' OR 'Out-of-pocket' OR 'Payments' OR 'Co-payments' OR 'Fee-for-service' OR 'Deductible' OR 'Expenditure' OR 'Financial burden' OR 'Financial contributions' OR 'Economic cost' AND;

2. 'Assisted Reproductive Technology (ART)' OR 'Medically Assisted Reproduction (MAR)' OR 'Invitro Fertilization (IVF)', 'Intrauterine Insemination (IUI)', 'Intracytoplasmic Sperm Injection (ICSI)', OR 'Infertility' OR 'Subfertility' OR 'Sterility' OR 'Fecundity' OR 'Subfecundity' OR 'Childlessness' AND;

3. 'High-income countries' OR 'Developed countries' OR 'Industrialized countries' OR 'North America' OR 'Western Europe' AND Specific country names AND Specific filters for high-income countries such as OECD countries [OVID], shall be applied: https://sites.google.com/a/york.ac.uk/issg-search-filters-resource/home/geography.

## Inclusion criteria

The review shall be guided by the PICOS framework [37, 38], a structured approach that optimizes search criteria and ensures core concepts guide the search. Below is the PICOS approach for this review.

a. Population: Individuals and couples seeking ART treatment in high-income countries.

b. Intervention: Assisted reproductive technology specifically IVF and ICSI.

c. Outcome Measures: The primary outcome is direct medical and direct non-medical costs associated with ART, while the secondary outcomes will include the presence and typology of factors influencing government and insurers' costs (such as ART financing, and eligibility criteria) and countries economic markers (such as average/minimum income, GNI per capita).

d. Study design: All study designs, including observational studies, experimental and quasi-experimental studies, and economic evaluations that report on the cost of ART in high-income countries will be eligible.

e. Timeframe: The search will be restricted to studies published between 1 January 2001 and 31 December 2024 in all languages. The extended timeframe will facilitate an exploration of the evolving nuances in the ART financing landscape over time.

## Exclusion criteria

a. Population: Individuals or couples seeking ART treatment outside HICs will be excluded.

b. Intervention: Non-IVF and non-ICSI procedures, such as intrauterine insemination (IUI), surrogacy, and fertility preservation outside the context of IVF/ICSI, will be excluded.

c. Outcome Measures: Studies that do not report on cost and studies that duplicate datasets or reference results from other included studies shall be excluded to avoid redundancy.

d. Study designs: Case reports, expert opinions, literature reviews from which original data is not provided, and editorials will not be included. Additionally, studies with ethical concerns or those that have been published and subsequently retracted will be deemed ineligible.

e. Time frame: Studies whose primary data pertains to periods before 2001, even if published after 2001, will be excluded.

## Study setting

The geographical focus of the review will be restricted to studies undertaken in HICs as defined by the World Bank (http://data.worldbank.org/country). The World Bank classifies countries according to their gross national income per capita into low, lower-middle, upper-middle and high income [39]. HICs are also referred to as developed countries.

## Definition of outcome and analysis parameters

a. *Direct medical costs*: refers to medical costs paid by patients per ART cycle to health providers irrespective of treatment outcomes, including pre-ART work-up, medical consultation, drugs, IVF procedures, hospital charges, laboratory costs, anesthesia fees, monitoring, counseling and administration charges.

b. *Direct non-medical costs*: refers to non-medical costs incurred by patients, such as transport, accommodation, and food.

c. *High-income countries*: refers to the countries defined to be above the World Bank threshold of GNI per capita according to 2023 estimates [39].

d. *ART financing*: refers to the government or third-party mechanism of ART funding and/or reimbursement as reported in the most current IFFS review [11].

e. *Regions*: refers to the classification of the world into WHO regions for administration and reporting purposes.

## Screening procedure

Two researchers will independently review the titles and abstracts of all identified studies. Studies will be included based on the above criteria. Any potential conflicts that arise on whether to include or exclude a study will be resolved through consensus and the reasons for

exclusion documented. If necessary, a third review author will be consulted to reach a consensus. In keeping with PRISMA guidelines for a systematic review [36], a flow diagram will be developed, showing the literature search, article selection, and final included studies. All the reviewers shall identify and agree on eligible studies for inclusion in the final evaluation.

During the initial screening, the titles and abstracts of non-English publications will be translated using Google Translate. Subsequently, eligible non-English papers will undergo complete translation using Google Translate. Additional translation methods will be explored if required and feasible. If necessary, the primary author of non-English publications will be contacted through email to validate the English translation of their work and the key findings.

## Data extraction

Two independent reviewers will extract relevant data from eligible studies using a predefined data extraction form. Data items will include study characteristics (author, year of publication, country of study), study objective, methodology (study design, sample size, study participants), types of infertility treatment, and cost-related information. In the event of uncertainty, the two reviewers will confer with each other and, if necessary, consult a third review author to reach a consensus.

Data from the included studies shall be complemented with data on the economic and healthcare system characteristics of the country. ART health financing data shall be extracted from the International Federation of Fertility Societies' Surveillance (IFFS) database [11]. While the macro-economic data, the GNI per capita (USD PPP), shall be extracted from the World Bank economic metrics [40], and the average income from the World Bank Shared Prosperity data [41].

## Risk of bias assessment

The studies' quality will be evaluated using the Integrated Quality Criteria for Review of Multiple Study Designs (ICROMS) tool, which allows for the assessment of multiple study designs [42]. ICROMS criteria encompass the assessment of randomized controlled trials, controlled before-and-after studies, interrupted time series, non-controlled before-and-after studies, cohort studies, and qualitative studies. The tool comprises a 'decision matrix' and a set of distinct quality standards associated with each study design, underpinned by a scoring system.

Two reviewers will systematically evaluate the methodological quality of the included studies, and their scores will be compared to ensure consistency. Any score discrepancies shall be discussed with a third reviewer.

## Data analysis and synthesis

Tables will be used to provide a summary of the data extracted from the studies assessed. A synthesis of findings will be conducted by summarizing and comparing the cost of ART (USD PPP) across countries and its association with ART financing, GNI per capita (USD PPP), and average annual income data. Given the expected heterogeneous nature of the outcomes of interest, this review shall largely focus on descriptive analysis.

A correlation matrix will be used to explore the associations between ART costs, GNI per capita, and ART financing. Countries shall be categorized according to the WHO classification of the world into six regions for purposes of administration and reporting. The regions include African Region (AFR), Region of the Americas (AMR), South-East Asian Region (SEAR), European Region (EUR), Eastern Mediterranean Region (EMR), and Western Pacific Region (WPR). Further analysis will include classification based on whether ART cycles are fully or partially funded, as well as the eligibility criteria of the respective countries. Overtime analysis

of changes in ART costs will be conducted once a clear cost pattern is established, providing insights into trends and factors driving cost fluctuations.

If data from the studies provides sufficient cost effect size estimates, meta-analysis will be performed by pooling cost data from similar studies.

## Discussion

This systematic review will provide a comprehensive overview of the cost of ART borne by patients in HICs, contributing to understanding the economic implications of fertility care for individuals, state funding institutions, and third-party insurance providers. The findings will be valuable for policymakers and healthcare providers to assess the performance and effectiveness of their respective healthcare systems in achieving universal health coverage. Also, policymakers would get comprehensive insights into good practices from other countries regarding access, and affordability of ART. Potential limitations of the review will be discussed, and recommendations for future research will be outlined.

A potential limitation of this review is the anticipated diverse ART policies across different countries and their evolution over time, which significantly influence the funding mechanisms for ART. Consequently, the ART costs reported in various studies may be influenced by the specific financing policies of each country, introducing complexities in the analysis and synthesis. Nonetheless, this review will aim to identify any correlations between the costs of ART and the prevailing financing mechanisms. Also, some countries may have advanced or otherwise revised their financing policies since the time when included studies were conducted, and in such cases, the review will endeavor to provide current context or discuss the implications of such changes on the reported findings.

## Supporting information

**S1 Checklist. PRISMA-P 2015 checklist.**
(DOCX)

**S1 File. Provisional search strings.**
(DOCX)

## Author Contributions

**Conceptualization:** Purity Njagi, Gitau Mburu.

**Data curation:** Purity Njagi.

**Methodology:** Purity Njagi, Wim Groot, Jelena Arsenijevic.

**Project administration:** Gitau Mburu, James Kiarie.

**Supervision:** Wim Groot, Jelena Arsenijevic, Gitau Mburu.

**Validation:** Georgina Chambers, Carlos Calhaz-Jorge.

**Writing – original draft:** Purity Njagi.

**Writing – review & editing:** Wim Groot, Jelena Arsenijevic, Gitau Mburu, Georgina Chambers, Carlos Calhaz-Jorge, James Kiarie.

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
