## [Decision Letter · Decision Letter 0]

7 Nov 2024

PONE-D-24-40151Financial cost of assisted reproductive technology for patients in high-income countries: A systematic review protocolPLOS ONE

Dear Dr. Njagi,

Thank you for submitting your manuscript to PLOS ONE. After careful consideration, we feel that it has merit but does not fully meet PLOS ONE’s publication criteria as it currently stands. Therefore, we invite you to submit a revised version of the manuscript that addresses the points raised during the review process.

We look forward to receiving your revised manuscript.

Kind regards,

Anil Gumber, Ph.D.

Academic Editor

PLOS ONE

“This work has received funding from the UNDP-UNFPA-UNICEF-WHO-World Bank Special Programme of Research, Development and Research Training in Human Reproduction (HRP), a cosponsored programme executed by the World Health Organization (WHO). Grant Number: APW-PO 203363893.”

Reviewers' comments:

Reviewer's Responses to Questions

**Comments to the Author**

1. Does the manuscript provide a valid rationale for the proposed study, with clearly identified and justified research questions?

Reviewer #1: Yes

Reviewer #2: Yes

Reviewer #3: Yes

2. Is the protocol technically sound and planned in a manner that will lead to a meaningful outcome and allow testing the stated hypotheses?

Reviewer #1: Yes

Reviewer #2: Yes

Reviewer #3: Yes

3. Is the methodology feasible and described in sufficient detail to allow the work to be replicable?

Reviewer #1: Yes

Reviewer #2: Yes

Reviewer #3: Yes

4. Have the authors described where all data underlying the findings will be made available when the study is complete?

Reviewer #1: Yes

Reviewer #2: Yes

Reviewer #3: Yes

5. Is the manuscript presented in an intelligible fashion and written in standard English?

Reviewer #1: Yes

Reviewer #2: Yes

Reviewer #3: No

6. Review Comments to the Author

You may also provide optional suggestions and comments to authors that they might find helpful in planning their study.

Reviewer #1: The authors present a protocol for a systematic review to evaluate the financial cost of assisted reproductive technology for patients in HIC. The protocol is well organized and written clearly. It is a highly relevant subject. I present below some minor comments to help strengthen the manuscript:

The objective of the abstract could be clearer. Is there a way for the author to be more specific, particularly regarding the concept of healthcare system characteristics: “to examine the ART costs paid by patients in HICs and the correlation with economic and healthcare system characteristics”

Spell out “ICROMS” in the abstract

I would invite the authors to provide an explanation of the choice of the timeframe and provide some reflections around the potential challenges of having such a large timeframe, particularly regarding different tendencies around ART access that may change with time.

For the screening procedure, will the authors use a software like Covidence?

For the inclusion and exclusion criteria section, I would invite the authors to divide that section into two parts – inclusion and exclusion criteria – and details to the exclusion criteria, such as Exclude studies that present the same dataset or results as another included study to avoid redundancy, and exclude studies with substantial ethical concerns.

In that same section, it will be important list which ART are included and excluded. For example: in vitro fertilization, intracytoplasmic sperm infection, artificial insemination, egg donation, sperm donation, surrogacy, fertility preservation (i.e., egg, sperm freezing), etc.

Provide a more thorough plan for data synthesis. Clearly explain how findings will be categorized, summarized, and compared. If a meta-analysis is not possible, discuss alternative approaches to synthesizing qualitative and quantitative data.

Consider conducting a pilot search to ensure that the proposed search strategy yields relevant and sufficient studies. This step would demonstrate feasibility and help fine-tune the process before full implementation.

Reviewer #2: Good greeting

I would like to thank the researchers for this idea of research because it studies an important problem in society, the language is clear ,correct and unambiguous

The search is excellent and deserves to be published

my regards

Reviewer #3: Financial cost of assisted reproductive technology for patients in high-income countries: A systematic review protocol

This study protocol provides the plan of the authors to write a review on the above topic. The authors aware the difference of the public and private IVF systems in some high-income countries (HIC), also payment Out of Pocket (OOP) as the consequences for the patients.

Currently the protocol indicates that the authors pay more attention on In Vitro Fertilisation. However, there are many other issues that can be included in this proposed review such as:

1. Cross Border Reproductive Care, because patients from HIC may go to LMIC to obtain ART treatment especially surrogation. The authors should be able to look at financial or cost distribution in each part such as legal cost, treatment for the couple and surrogate etc.

2. Gamete or embryo donation this may occur in both HIC and LMIC.

3. Social fertility preservation.

4. Consider the role of children in family that motivates people in HIC to have their own child.

It is suggested that authors discusss those issue since this will reveal the main reason of couple in HIC to have their own child.

7. PLOS authors have the option to publish the peer review history of their article (what does this mean?). If published, this will include your full peer review and any attached files.

Reviewer #1: No

Reviewer #2: **Yes: **Sawsan S. Hamzah

Reviewer #3: **Yes: **Mulyoto Pangestu

---

## [Author Response · Author response to Decision Letter 0]

21 Dec 2024

RE: PONE-D-24-40151

Financial cost of assisted reproductive technology for patients in high-income countries: A systematic review protocol

Response to Reviewers

Reviewer #1: The authors present a protocol for a systematic review to evaluate the financial cost of assisted reproductive technology for patients in HIC. The protocol is well organized and written clearly. It is a highly relevant subject. I present below some minor comments to help strengthen the manuscript:

The objective of the abstract could be clearer. Is there a way for the author to be more specific, particularly regarding the concept of healthcare system characteristics: “to examine the ART costs paid by patients in HICs and the correlation with economic and healthcare system characteristics”

Response: We agree with this comment and recognize that "healthcare system characteristics" is a broad term, yet we specifically refer to a few factors related to ART regulations. Consequently, we have revised the statement to be more precise, with the changes reflected in both the abstract and the main body of the text. (Page 7: line 143 & 148) 

“to examine the ART costs paid by patients in HICs and the correlation with economic indicators and ART regulatory frameworks”. 

Under the specific objective we have expanded as follows (Page 8: Line 157) 

“Examine the correlation between the ART costs borne by patients across the HICs with ART regulatory frameworks, including financing and eligibility criteria.”

Spell out “ICROMS” in the abstract

Response: The full words are now provided in the abstract

“Integrated Quality Criteria for Review of Multiple Study Designs (ICROMS)”

I would invite the authors to provide an explanation of the choice of the timeframe and provide some reflections around the potential challenges of having such a large timeframe, particularly regarding different tendencies around ART access that may change with time.

Response: Thank you for this comment. We agree that analyzing the data for an extended period of time may present challenges due to changes in regulations over time, as well as increases in costs driven by advancements in treatment and inflation. However, we have chosen this timeframe to explore the possible nuances of these differences over time.

For clarity, we have provided an explanation of the timeframe as follows: “The extended timeframe will facilitate an exploration of the evolving nuances in the ART financing landscape over time.” (Page 10: Line 223-225)

Under limitation, we had included the difference in policies, and thus we have added the time element (Page 14: Line 322). “A potential limitation of this review is the anticipated diverse ART policies across different countries and their evolution over time, which significantly influence the funding mechanisms for ART.”

Additionally, this review builds upon a similar review of LMICs that examined the same timeframe, and we aim to maintain consistency in the timeframe for comparison purposes. Given we shall finalize the search early 2025, we have revised the time to 2001-2024 (Page 10: Line 225). 

For the screening procedure, will the authors use software like Covidence?

Response: Thank you for this suggestion. We agree that software like Covidence would be useful for a systematic review, however we have decided to proceed with a “manual screening” approach supported by a reference management software (Zotero). This decision is informed by the heterogeneity of the studies and the variability in how cost data is reported across them, as well as lessons learned from a previous similar review conducted in LMICs.

For the inclusion and exclusion criteria section, I would invite the authors to divide that section into two parts—inclusion and exclusion criteria—and details to the exclusion criteria, such as exclude studies that present the same dataset or results as another included study to avoid redundancy and exclude studies with substantial ethical concerns.

Response: We agree with this comment and have thus separated the inclusion and exclusion criteria; and expanded on the exclusion criteria as follows (Page 10 & 11: line 227 -240)

• Population: Individuals or couples seeking ART treatment outside HICs will be excluded.

• Intervention: Non-IVF and non-ICSI procedures, such as intrauterine insemination (IUI), surrogacy, and fertility preservation outside the context of IVF/ICSI, will be excluded.

• Outcome Measures: Studies that do not report on cost and studies that duplicate datasets or reference results from other included studies shall be excluded to avoid redundancy.

• Study designs: Case reports, expert opinions, literature reviews from which original data is not provided, and editorials will not be included. Additionally, studies with ethical concerns or those that have been published and subsequently retracted will be deemed ineligible.

• Time frame: Studies whose primary data pertains to periods before 2001, even if published after 2001, will be excluded.

In that same section, it will be important to list which ART are included and excluded. For example: in vitro fertilization, intracytoplasmic sperm infection, artificial insemination, egg donation, sperm donation, surrogacy, fertility preservation (i.e., egg, sperm freezing), etc.

Response: For clarity we have enhanced the inclusion and exclusion criteria as follows: 

Under inclusion criteria (Page 10: Line 211): “Assisted reproductive technology, specifically IVF and ICSI” 

Under Exclusion criteria (Page 11: Line 230-232), Non-IVF and non-ICSI procedures, such as intrauterine insemination (IUI), surrogacy, and fertility preservation outside the context of IVF/ICSI, will be excluded.

Provide a more thorough plan for data synthesis. Clearly explain how findings will be categorized, summarized, and compared. If a meta-analysis is not possible, discuss alternative approaches to synthesizing qualitative and quantitative data.

Response: We have enhanced the data synthesis section with the analysis that shall be conducted. Depending on the trends/patterns of the data extracted, additional analysis may be conducted (Page 13 & 14: Line 302 – 310)

“A correlation matrix will be used to explore the associations between ART costs, GNI per capita, and ART financing. Countries shall be categorized according to the WHO classification of the world into six regions for purposes of administration and reporting. The regions include African Region (AFR), Region of the Americas (AMR), South-East Asian Region (SEAR), European Region (EUR), Eastern Mediterranean Region (EMR), and Western Pacific Region (WPR). Further analysis will include classification based on whether ART cycles are fully or partially funded, as well as the eligibility criteria of the respective countries. Overtime analysis of changes in ART costs will be conducted once a clear cost pattern is established, providing insights into trends and factors driving cost fluctuations.”

Consider conducting a pilot search to ensure that the proposed search strategy yields relevant and sufficient studies. This step would demonstrate feasibility and help fine-tune the process before full implementation.

Response: We had considered a pilot search, which we are undertaking with the assistance of an information specialist (a librarian) from the University Library to strengthen the search strategy. We have enhanced the search strategy wording to clarify that we will conduct a pilot (Page 8: Line 173)

“A systematic literature search strategy will be developed and implemented in collaboration with an academic librarian who will assist in developing search strings, refining search terms, optimizing and piloting the search strategy to ensure a comprehensive search of the relevant literature.”

Reviewer #2: Good greeting

I would like to thank the researchers for this idea of research because it studies an important problem in society, the language is clear, correct and unambiguous

The search is excellent and deserves to be published

my regards

Response: We greatly appreciate your review.

Reviewer #3: Financial cost of assisted reproductive technology for patients in high-income countries: A systematic review protocol

We greatly appreciate your review and comments.

This study protocol provides the plan of the authors to write a review on the above topic. The authors aware the difference of the public and private IVF systems in some high-income countries (HIC), also payment Out of Pocket (OOP) as the consequences for the patients.

Currently the protocol indicates that the authors pay more attention on In Vitro Fertilisation. However, there are many other issues that can be included in this proposed review such as:

1. Cross Border Reproductive Care, because patients from HIC may go to LMIC to obtain ART treatment especially surrogation. The authors should be able to look at financial or cost distribution in each part such as legal cost, treatment for the couple and surrogate etc.

2. Gamete or embryo donation this may occur in both HIC and LMIC.

3. Social fertility preservation.

4. Consider the role of children in family that motivates people in HIC to have their own child.

It is suggested that authors discuss those issue since this will reveal the main reason of couple in HIC to have their own child.

Response: Thank you for this comment, which highlights the importance of considering additional factors in determining ART costs.

Regarding the issues raised, we have addressed them as follows:

1. Cross Border Reproductive Care (CBRC) in the background (Page 6: Line 106-110) as a means through which individuals seek more affordable ART prices in other countries. 

2. Any cost associated with CBRC will be included, provided it meets the inclusion criteria; meaning it is associated with IVF or ICSI costs in HICs. 

3. Gamete and embryo donation will be considered within the context of IVF or ICSI.

4. Fertility preservation methods, such as embryo freezing, will be considered within the context of IVF purposes.

We have expanded the inclusion and exclusion criteria to specify ART procedures that will be eligible, based on your comment and that of another reviewer. 

Under inclusion criteria (Page 10: Line 211): Intervention: “Assisted reproductive technology, specifically IVF and ICSI.” 

Under Exclusion criteria (Page 11: Line 230), Non-IVF and non-ICSI procedures, such as intrauterine insemination (IUI), surrogacy, and fertility preservation outside the context of IVF/ICSI, will be excluded.

We recognize the numerous factors that influence couples and individuals in their desire to have children and how these factors affect their willingness to pay for ART. As such we shall incorporate the contributions of these factors into the discussion, as reflected in the findings of the eligible studies that will be analyzed under this review.

---

## [Decision Letter · Decision Letter 1]

22 Jan 2025

Financial cost of assisted reproductive technology for patients in high-income countries: A systematic review protocol

PONE-D-24-40151R1

Dear Dr. Njagi,

We’re pleased to inform you that your manuscript has been judged scientifically suitable for publication and will be formally accepted for publication once it meets all outstanding technical requirements.

Kind regards,

Patrick Goymer

Staff Editor

PLOS ONE

Additional Editor Comments (optional):

Reviewers' comments:

Reviewer's Responses to Questions

**Comments to the Author**

1. Does the manuscript provide a valid rationale for the proposed study, with clearly identified and justified research questions?

Reviewer #1: Yes

Reviewer #2: Yes

Reviewer #3: Yes

2. Is the protocol technically sound and planned in a manner that will lead to a meaningful outcome and allow testing the stated hypotheses?

Reviewer #1: Yes

Reviewer #2: Yes

Reviewer #3: Yes

3. Is the methodology feasible and described in sufficient detail to allow the work to be replicable?

Reviewer #1: Yes

Reviewer #2: Yes

Reviewer #3: Yes

4. Have the authors described where all data underlying the findings will be made available when the study is complete?

Reviewer #1: Yes

Reviewer #2: Yes

Reviewer #3: Yes

5. Is the manuscript presented in an intelligible fashion and written in standard English?

Reviewer #1: Yes

Reviewer #2: Yes

Reviewer #3: Yes

6. Review Comments to the Author

You may also provide optional suggestions and comments to authors that they might find helpful in planning their study.

Reviewer #1: Thank you for your revisions, the protocol is now much more robust and clearer. Good luck with the actual review.

Reviewer #2: I would like to thank the researchers for this research idea because it studies an important problem in society, and the language is clear, correct and unambiguous ,The research is excellent and Especially after the author made the corrections required by the reviewers deserves to be published

My Regards

Reviewer #3: I would like to thanks to the authors who already made all the correction and adding further information as requested by reviewers.

7. PLOS authors have the option to publish the peer review history of their article (what does this mean?). If published, this will include your full peer review and any attached files.

Reviewer #1: No

Reviewer #2: **Yes: **Sawsan S. Hamzah

Reviewer #3: **Yes: **Mulyoto Pangestu

---

## [Editor Report · Acceptance letter]

30 Jan 2025

PONE-D-24-40151R1 

PLOS ONE

Dear Dr. Njagi, 

I'm pleased to inform you that your manuscript has been deemed suitable for publication in PLOS ONE. Congratulations! Your manuscript is now being handed over to our production team.

Kind regards, 

on behalf of

Dr Patrick Goymer 

Staff Editor

PLOS ONE